# Modeling cell biological features of meiotic chromosome pairing to study interlock resolution

Erik J. Navarro[1], Wallace F. Marshall[2‡], Jennifer C. Fung[1‡*]

**1** Department of Obstetrics, Gynecology and Reproductive Sciences and Center of Reproductive Sciences, University of California, San Francisco, California, United States of America, **2** Department of Biochemistry and Biophysics, University of California, San Francisco, California, United States of America

‡ These authors are joint senior authors on this work.
* jennifer.fung@ucsf.edu

**Data Availability Statement:** Lammps input script, data, and codes are available at: https://github.com/eriknavarro/meiotic-pairing-model.

## Abstract

During meiosis, homologous chromosomes become associated side by side in a process known as homologous chromosome pairing. Pairing requires long range chromosome motion through a nucleus that is full of other chromosomes. It remains unclear how the cell manages to align each pair of chromosomes quickly while mitigating and resolving interlocks. Here, we use a coarse-grained molecular dynamics model to investigate how specific features of meiosis, including motor-driven telomere motion, nuclear envelope interactions, and increased nuclear size, affect the rate of pairing and the mitigation/resolution of interlocks. By creating in silico versions of three yeast strains and comparing the results of our model to experimental data, we find that a more distributed placement of pairing sites along the chromosome is necessary to replicate experimental findings. Active motion of the telomeric ends speeds up pairing only if binding sites are spread along the chromosome length. Adding a meiotic bouquet significantly speeds up pairing but does not significantly change the number of interlocks. An increase in nuclear size slows down pairing while greatly reducing the number of interlocks. Interestingly, active forces increase the number of interlocks, which raises the question: How do these interlocks resolve? Our model gives us detailed movies of interlock resolution events which we then analyze to build a step-by-step recipe for interlock resolution. In our model, interlocks must first translocate to the ends, where they are held in a quasi-stable state by a large number of paired sites on one side. To completely resolve an interlock, the telomeres of the involved chromosomes must come in close proximity so that the cooperativity of pairing coupled with random motion causes the telomeres to unwind. Together our results indicate that computational modeling of homolog pairing provides insight into the specific cell biological changes that occur during meiosis.

## Author summary

Early in meiosis, homologous chromosomes must find each other within the crowded nuclear space and become aligned along their entire length in a process known as

**Funding:** This study was funded by the National Institute of Health 5R01GM137126 (JCF), National Institute of Health R35 GM130327 (WFM) (https://www.nih.gov/) and National Science Foundataion (https://www.nsf.gov/) DBI-1548297 (WFM, JCF). The funders had no role in study design, data collection and analysis, decision to publish, or preparation of the manuscript.

**Competing interests:** The authors have declared that no competing interests exist.

homologous chromosome pairing. It remains unclear how the cell manages to align each pair of chromosomes quickly while mitigating and resolving interlocks. Here, we study this process by using a computational model. Our model attempts to capture the large-scale cell biological picture of meiotic pairing including the random initial 3D search, active motion of the chromosome ends, and meiosis specific constraints such as telomere attachment to the nuclear envelope. We use our model to study how these different features of meiosis affect the rate of pairing and the mitigation/resolution of interlocks. Importantly, our model gives us detailed movies of interlock resolution events, which we then analyze to build a step-by-step recipe for interlock resolution. We believe computational modeling of homolog pairing provides valuable insight into this complex biological process.

## Introduction

Meiotic chromosome pairing is a necessary first step for meiotic recombination and segregation. At the start of meiosis, homologous chromosomes must locate each other in the crowded nuclear space then become aligned along their entire length in a process known as homologous chromosome pairing (**Fig 1A**). Once sections of the chromosomes are paired, a complex of proteins which forms the synaptonemal complex assembles between paired regions of the chromosomes and holds them together [1]. This tight end-to-end alignment after the completion of synapsis allows for recombination, in which information is exchanged between the maternal and paternal homologs and generates crossovers which promote faithful chromosome segregation during meiosis I.

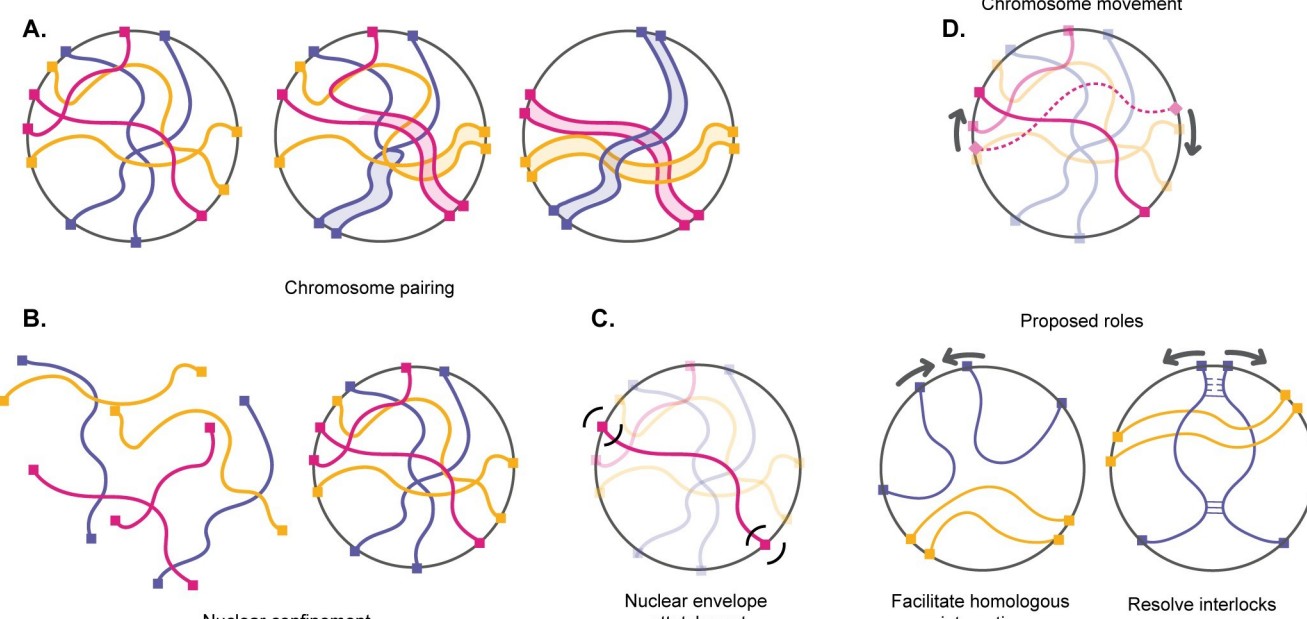

**Fig 1. The physical context of meiotic pairing.** (A) Cartoon schematic of chromosome pairing (B) Cartoon schematic of nuclear confinement (C) Cartoon schematic of telomere attachment to the nuclear envelope (D) chromosome movement and two proposed roles: To increase collisions between homologous loci, or to resolve interlocks by breaking apart problematic pairing interactions.

Despite the importance of pairing in the proliferation of sexually reproducing organisms, very little is known about chromosome pairing as a physical process. Unlike many molecular recognition processes which occur at a nanometer length scale, meiotic pairing requires motion on the micron length scale, and thus faces special challenges owing to the large size and dense packing of the chromosomes within the nucleus (**Fig 1B**). The homology search process thus involves not only a huge number of individual homology-assessment interactions but also a physical challenge of moving such large macromolecular structures over long distances in a densely tangled environment.

Several previous computational models have been described to represent meiotic homolog pairing [2–7]. These models have, in general, ignored the possibility of entanglement, by assuming that chromosomes are able to pass through one another. Such a "phantom polymer" assumption has been supported on theoretical grounds for mitotic chromosomes based on estimates of topoisomerase II (topo II) mediated strand passage rates [8]. However, there are several concerns about making such an assumption for meiotic chromosomes. First, in contrast to interphase chromosomes, consisting of a single DNA double helix, each meiotic homolog consists of two sister chromatids, closely cohering together. In order to pass one meiotic homolog through another, it would be necessary to pass a pair of double helixes through another pair of double helixes, and this is not a known activity of topo II. While it is known that topo II is upregulated during meiosis [9] and that mutations in topo II lead to meiotic defects [10,11], there is, to our knowledge, no direct demonstration that topo II can pass one meiotic chromosome through another. Genetic experiments show that topo II mutants have many different effects on meiosis, for example affecting recombination itself, [11,12], hence neither the upregulation of topo II in meiosis, nor the defects that result from topo II mutations, necessarily show a specific role in passing chromosomes through each other, and may instead reflect other roles of topo II for example in chromatin loop organization. Moreover, in many organisms, by the leptotene stage in meiosis when chromosomes undergo pairing, a protein-based axial element has already assembled on each homolog [13–16]. The proteins that make up the axial element self-assemble into continuous protein filaments [17,18]. In cases where the axial element has already assembled by the time pairing takes place, it is obviously not possible for topo II to catalyze the passage of these protein-based linear elements. Finally, we note that the persistence of interlocks through pachytene [13,19] demonstrates the inability of the chromosomes to simply pass through one another. Interlocks are topological constraints that prevent pairing completion; they occur when a chromosome becomes trapped between another pair of chromosomes that have at least partially paired on each side. While the term interlocks is usually used in the context of meiosis, recent work has raised the idea that interphase chromosomes may also be partially knotted [20–22].

One potential solution to the problem of moving chromosomes through a tangled mass of other chromosomes would be reptation—the snake-like slithering motion of a polymer through a network of other polymers. However, reptation is extremely slow compared to free diffusion [23]. In principle, molecular motors could provide a driving force to accelerate chromosome motion (**Fig 1C**). Indeed, meiotic chromosomes are subject to forces generated by myosin or dynein motors, depending on the species, which are able to exert forces on telomeres attached to the nuclear envelope (NE). This attachment is provided by SUN/KASH proteins that span the entire NE [24] so that the chromosome ends are pulled back and forth in the plane of the nuclear membrane by motor proteins outside the nucleus [25–29]. It has been hypothesized that this active motion aids the pairing process either by increasing collisions between homologous loci, by testing homology, and/or by helping to resolve interlocks that occur as part of the pairing process (**Fig 1D**) [4,27,30,31]. Mutations in telomere coupling to the actin cytoskeleton in yeast lead to reduced encounters between homologous loci [32].

Although reduction in motion leads to delays in completing meiosis [33,34], it can also cause an increase in total crossover number [30,35]. In fission yeast, when active motion is transiently stopped, initial pairing is slower, but then hyperstable pairing associations form that appear to involve unresolvable recombination events which eventually block proper chromosome segregation [36]. Thus, while rapid telomere motions clearly affect the meiotic process, their exact role remains unclear.

Clearly, there are still many unanswered questions about how meiotic chromosome pairing occurs. Part of the difficulty of studying this process experimentally using *in vivo* fluorescence microscopy comes from the fact that even in an organism with relatively short chromosomes such as S. *cerevisiae*, pairing is still a slow step in meiosis, taking hours to complete. At these timescales, photobleaching & phototoxicity are problematic, even when using very low-intensity light. The difficulty of studying this process experimentally coupled with the fact that very little is known about pairing as a physical process, motivated us to study this process using a computational model. Our main goal when building our model was to capture the large-scale topology of the pairing problem, not necessarily the exact molecular details of pairing/unpairing.

## Methods

We modeled meiotic chromosome dynamics using a coarse-grained molecular dynamics model. In our model framework, each chromosome is a list of nodes representing beads connected by springs, with each node subjected to Langevin random forces. To solve the system numerically, we use the thermostat described in [37] to model the interaction of each node with an implicit solvent, and NVE integration to update the velocity and position of each node at every timestep. The force on each node then has the form:

$$F = F_r + F_f + F_s + F_{lj} \tag{1}$$

$$F_r \propto \sqrt{\frac{k_b T m}{d_t \zeta}} \tag{2}$$

$$F_f = -\left(\frac{m}{\zeta}\right) * v \tag{3}$$

$$F_s = -\nabla(k_s(r_{(i,i+1)} - r_0)^2) \tag{4}$$

$$F_{lj} = -\nabla\left(4\varepsilon\left(\left(\frac{\sigma}{r_{(i,j)}}\right)^{12} - \left(\frac{\sigma}{r_{(i,j)}}\right)^6\right)\right), r_{(i,j)} < r_c \tag{5}$$

Where $F_r$ is the root mean square magnitude of the Langevin random force due to interactions with the solvent, $k_b$ is the Boltzmann constant, T is the temperature, m is the mass of a node, $d_t$ is the length of a timestep, and $\zeta$ is the damping time. $F_f$ is the frictional force term, which is proportional to the velocity of the node, $F_s$ is a harmonic spring potential (which has a corresponding spring constant) which provides the restoring spring force between adjacent nodes and $F_{lj}$ is the Lennard Jones force (which has two coefficients: $\varepsilon$ with units of energy, and $\sigma$ with units of distance) to keep nodes from overlapping in space. The LJ potential has both a repulsive and an attractive term, but the cutoff radius is set to be the minimum of the potential so that nodes only experience the repulsive portion. Together, these equations

simulate polymers undergoing Brownian motion in a viscous implicit solvent while including excluded volume interactions to prevent strand passage. All simulations are carried out using the molecular dynamics package LAMMPS [38] and the equations describing the forces have been included here for completeness.

At the beginning of the simulation, each polymer chain representing a chromosome is initialized on a cubic lattice. All nodes except for the terminal nodes start on this lattice. The terminal nodes begin on the nuclear membrane. We note that this initial packing on a lattice is simply to ensure that nodes do not begin the simulation overlapped. During all subsequent simulation and equilibration steps the nodes are free to move in 3D continuous space. The polymer then goes through two separate phases of equilibration. During the first equilibration, the maximum distance each node can move in a single timestep is capped to a distance 1/10 the size of the node. This allows the polymers to physically separate in 3D space and slowly leave the tightly packed initial lattice configuration. During the second equilibration, the distance each node can move in a single timestep is now uncapped, which serves to randomly orient the polymers in space. The length of the second equilibration was chosen such that subsequent measurements of the average radius of gyration matched theoretical predictions for the radius of gyration in a random chain model. In other words, we have equilibrated long enough that the polymer has "forgotten" it was initially packed in a lattice. The end of the second equilibration marks our initial timestep ($t = 0$), and all physical measurements begin from this point. These initialization/equilibration steps give us a starting configuration that represents two pairs (four chromosomes total) of homologous chromosomes with random relative orientations, confined inside of a nuclear sphere.

Unless otherwise stated, all simulations run for ten million timesteps. This corresponds to about 1.4 hours in real time, which is of the right order for pairing in budding yeast (hours).

To constrain the model chromosomes to a nucleus-like region, we use an indenter to keep all nodes in the simulation confined to a sphere with a chosen radius. An indenter is a built-in function in LAMMPS whose purpose is to act as a constraining wall (in our case a spherical wall) within a simulation. The indenter exerts a force on all nodes:

$$F(r) = -k_{nuc}(s - R)^2, s < R \tag{6}$$

Where $k_{nuc}$ is a force constant, s is the distance from the node to the center of the indenter, and R is the indenter radius. To keep the ends of the chromosomes confined to the nuclear surface, we use the constraint algorithm described in [39], which ensures that the net force on each telomeric node is always perpendicular to the surface of the constraining sphere. This constraint is managed by a built-in LAMMPS function called nve/manifold/rattle. Rapid telomere movements are modeled as randomly oriented large (relative to thermal forces) persistent forces that drag the telomeric nodes along the nuclear surface. To achieve this, the simulation is essentially divided into 2000 timestep chunks. For every chunk of the simulation, each telomeric node has a 25% chance of experiencing an active pull. For each telomere experiencing an active pull, a random straight-line direction is chosen. A force is applied in that direction for the entire chunk of time (2000 timesteps). For simulations involving a meiotic bouquet, an additional straight-line acceleration is exerted on the telomeric nodes that tend to keep these nodes in a subregion of the nuclear surface. The bouquet is modeled as a weak constant acceleration (force/mass) that drives the nodes toward a single point on the nuclear surface. The size of the bouquet can be modulated by adjusting the magnitude of this acceleration.

For the purpose of pairing, every node that makes up a chromosome is enumerated such that the first node along a polymer chain is node one, and the second node is node two, etc.

Pairing occurs when a node on one polymer chain comes within a specified capture distance of the corresponding node on the homologous chain. Pairing is modeled by binding two nodes together with a harmonic spring. The creation of this spring occurs with a specified probability, but only when two nodes come within the specified capture distance. The pairing of two nodes is reversible and falls apart with a specified dissociation probability.

### Parameter choice

The intent of our model is not to represent the chromosomes of any species in exact detail, instead, we take a toy model approach to capture the most important physical aspects associated with meiotic pairing: 1) Chromosomes are long linear polymers. 2) Chromosomes are confined to a nuclear volume. 3) The ends of each chromosome are attached to the nuclear envelope where they experience forces that drag them along the nuclear surface. 4) Each chromosome needs to find its homolog partner and become aligned from end to end. 5) Pairing needs to be completed while avoiding or resolving topological interlocks.

Taken together this represents a unique and complex physical situation that lends itself well to this type of model. Our parameter choices thus reflect order of magnitude estimates for the size/density parameters in the simulations, while energy and time scales are chosen such that the diffusion coefficient of our simulated polymers approximately matches experimental measurements of the diffusion coefficient of chromatin.

### Nuclear radius

To obtain order of magnitude estimates for size and length scales, we use *Saccharomyces cerevisiae* (budding yeast) as a representative organism since it is a widely used model organism in experimental studies of meiotic pairing. The radius of a meiotic yeast nucleus is approximately 1.2 microns. We take the fundamental unit of length in our simulations to be 100 nm, which makes the nuclear radius equal to 12 of these fundamental units.

### Polymer node size

The persistence length of interphase yeast chromatin has been measured to be approximately 50 nm [40–42]. While the chromosomes condense significantly throughout the course of meiosis, pairing begins very early in prophase while the chromosomes are largely uncondensed. In standard random chain polymer models, the persistence length is equal to half the Kuhn segment length of the chain. We therefore take the diameter of a node, which represents the segment length of the random chain model, in our simulation to be 100 nm, which corresponds to a length of 1 in our model units.

### Chromosome length

After the formation of the synaptonemal complex, the largest yeast chromosome has been observed to have a length of about 3 microns [43]. Since the synaptonemal complex forms after pairing, and thus after significant chromosomal condensation, we choose a slightly larger 10 microns as our chromosome length, which means each chromosome is made up of 100 nodes in our model framework.

### Chromosome number

We simulate four chromosomes (two pairs of homologs) in order to be able to investigate the potential for interlocks between non-homologous chromosomes. With our assumptions of

chromosome length, polymer node size, and nuclear volume, this results in a volume fraction of chromatin of 5%, which is comparable to prior estimates [44].

## Capture distance

Since our simulations include excluded volume effects that will prevent the nodes from overlapping, we take the capture distance for pairing to be the diameter of a node (100 nm). This means that in order to pair, two homologous nodes must physically touch.

## Spring constant

Our primary concern when choosing a spring constant is to set it high enough to prevent strand passage. To achieve this, we set our spring constant to 100 in our model units, which corresponds to a spring constant of 0.0417 pN/nm.

## Pairing parameters

In [6], a single locus was tagged on both homologous chromosomes and tracked over time. In that study, the spots corresponding to homologous loci reached a level of colocalization of approximately ~60%. Our pairing/unpairing parameters were chosen so that approximately ~55% of the pairing nodes are paired after a full run. This corresponded to a pairing probability of 85% (if within the cutoff radius), and an unpairing probability of 0.00125%

## Damping time & remperature

When choosing values for the temperature and damping time, there are two timescales of interest. The first is the timescale at which the momentum relaxes, this timescale is set in LAMMPS directly:

$$\tau_m = \zeta \tag{7}$$

$\zeta$ is the damping time that appears in Eq 2 & Eq 3. It sets the viscosity in the simulation (smaller damping time means larger viscosity). There is also a timescale (called the Brownian timescale) in which the particle diffuses a distance comparable to its own size

$$\tau_{bd} = \frac{\sigma^2 m}{\zeta K_b T} \tag{8}$$

The Brownian regime of Langevin dynamics occurs when:

$$\tau_{bd} \ll \tau_m \tag{9}$$

This is the regime accurate for high viscosities, which chromatin diffusion in the nucleus is thought to occupy. Our primary concern when choosing the temperature value and damping time is to ensure we remain within this regime. To do this, we set the damping time equal to 0.1 (5 milliseconds in physical units) and the temperature to 1 (~300 Kelvin in physical units), which gives us a Brownian timescale that is many times larger than the momentum dissipation timescale. We note that these parameter choices also give us a reasonable measurement of the simulated chromatin diffusion coefficient. We calculate the mean squared displacement (MSD) of the ten centermost nodes in each polymer and plot the MSD vs time. We then fit a line to the linear region of the resulting plot and obtain a diffusion coefficient of ~600nm^2/s (**S1 Fig**), which is comparable to the experimentally measured diffusion coefficient of ~500nm^2/s for yeast chromatin measured *in vivo* [45].

## Telomeric velocities

In the WT strain of budding yeast, the telomeres experience rapid pulls due to coupling with motor proteins outside the nucleus. These pulls have been experimentally measured to peak at approximately ~0.6 μm/s in most cells [27]. In the *csm4Δ* strain (which lacks the rapid telomere movements but telomeres remain on the envelope), the telomeres have been measured to have an average velocity of about ~0.05 μm/s [27]. In the *ndj1Δ* strain, the ends become detached from the nuclear envelope and thus are free to diffuse through the nucleus [46].

To model these three strains in silico, we set our telomeric velocities accordingly. In our simulated WT strain, the telomeres undergo rapid movement along the nuclear envelope which peaks at approximately ~0.6 μm/s (**S2 Fig**). The telomeres in our simulated *csm4Δ* strain lack these rapid movements, and have an average velocity of around ~0.05 μm/s. We model the *ndj1Δ* strain by removing both the constraint that keeps the telomeres attached to the nuclear envelope, and the rapid telomere movements.

## Plots & figures

Our results are generated as follows.

**Percent of paired nodes vs time.**   These plots were created by averaging the percentage of paired nodes at each timestep and plotting the results as a function of time. Error bars represent the 95% confidence interval on the mean value for each timepoint. N = 100 simulations were used to create each plot. Timepoints with non-overlapping confidence intervals are deemed to be significantly different.

**Percent of simulations with interlocks.**   These plots were created from a binary experiment in which (after the end of the simulation) each simulation is assessed for the existence of an interlock. This assessment is done automatically after the end of each simulation by forcing all nodes to pair. Simulations that contain interlocks will fail to finish pairing and are then counted. Error bars are estimated using the 95% binomial proportion confidence interval. Bars with non-overlapping confidence intervals are deemed to be statistically significant. N = 100 simulations were used to create each plot, except for one case, where it was necessary to use N = 200 simulations to establish statistical significance. Statistical significance or lack thereof is determined by using a two proportion z-test. In the case that error bars overlap and are not statistically significant, we have labeled the plots accordingly.

**Geometry of interlock resolution.**   For every interlock that forms and eventually resolves, our model gives a timestep-by-timestep molecular dynamics trajectory that includes the velocity and position of every single node in the simulation. We can then use tools such as OVITO [47] to visualize and animate these trajectories into interactive molecular movies where we can pan, zoom, rotate, hide nodes, etc. This gives us a powerful tool for studying the resolution of interlocks because otherwise, the highly dense, confined, thermally driven polymer system would look like a tangled mess.

**Timescales of interlock resolution steps.**   To estimate the timescales of the different interlock resolution steps, we visually analyzed the trajectories of fifteen interlock resolution events using the software program OVITO [47]. For each interlock resolution event, we noted the time it took to complete the three steps in interlock resolution. While we note that there is some ambiguity in calling each of these steps visually, we tried our best to adhere to these rules when picking the timestep at which each of these events occurs. Time to interlock migration: This is the timestep immediately preceding the interlock resolution event in which the interlock first gets within 10 nodes of the polymer endpoints. Time to telomeric unpairing: Following interlock migration, this is the time it takes for the telomeric regions of the involved polymers to fully unpair. Time to diffusive unwinding: Following telomeric unpairing, this is

the time it takes for both pairs of homologous chromosomes to appear disentangled from each other such that a recurrence of an interlock is unlikely.

**Primitive Path Analysis (PPA).** We used primitive path analysis to count the number of entanglements along our polymer chains. Primitive path analysis [48] is a computational method which forces a polymer to condense to its primitive path. The primitive path can be defined as the shortest path through all the entanglements. For an unentangled polymer, the primitive path will be the shortest straight line that connects both polymer endpoints. For an entangled polymer however, the primitive path will have "kinks" at places of entanglement due to excluded volume interactions. These "kinks" can be counted and directly represent the number of entanglements along a polymer chain. We ran the primitive path algorithm on 15 simulations. For each simulation we equilibrate the four polymers as usual. At the end of the equilibration, we designate one of the four polymers as the polymer of interest (the PPA algorithm requires you to choose a polymer of interest), run PPA algorithm on that polymer, visualize the primitive path in OVITO [47], and then manually count the number of entanglements along that polymer chain.

**Kymographs of individual pairing trajectories.** To highlight the phenomenon of zippering (or lack thereof), we have plotted the distance between corresponding nodes on homologous chains as a heatmap through time as it was done in [4,5]. The y-axis is the number of the nodes along the polymer chain (1–100), while the x-axis is time. The color of each bin represents the distance between homologous nodes. Fast zippering in these plots will appear as a sudden, near vertical shift to the color blue.

## Results

### Pairing sites that are spread out along the length of the chromosomes correlate best with experimental data from wild-type, *ndj1Δ* and *csm4Δ* yeast strains

In order to gain insights into the possible roles of large-scale mechanical and organizational features of meiotic chromosomes on the process of pairing in a densely entangled nucleus, we carried out molecular dynamics simulations (**Fig 2**). Our simulation framework incorporated parameter values estimated from existing literature as detailed in Methods and used the LAMMPS platform [38] to model chromosome motion driven by Langevin random forces. In this model, chromosome polymers are represented as bead-spring chains, with the beads corresponding to polymer domains on the order of one persistence length. The nuclear envelope (NE) is represented as a confining sphere. To represent telomere attachment to the nuclear envelope, the initial and final beads on each chain are constrained to be on the nuclear sphere. To represent rapid telomere motion (RTM), large (relative to thermal) forces are applied to the initial and final nodes, as detailed in Methods. To represent entanglement, excluded volume interactions are modeled to prevent passage of polymers through each other.

Using this framework, we carried out simulations in the presence of RTMs, as well as in two cases designed to represent well-studied mutants. In *cms4Δ* mutants, telomeres are attached to the NE but are not subject to cytoskeletal forces [30]. We model this mutant by eliminating the simulated RTMs at the terminal nodes of the chains, such that all nodes of the chain are subject to the same standard thermal Langevin random force. In *ndj1Δ* mutants, telomeres are detached from the NE [46], a situation that we model by removing the constraint for the nodes to be anchored on the NE, as well as eliminating the RTMs. We then compared these three simulations in order to determine how the presence of RTMs and/or NE attachment might influence the pairing process. As shown in (**Fig 3A**), the result was that homolog pairing in both simulated mutants progressed with approximately the same kinetics as

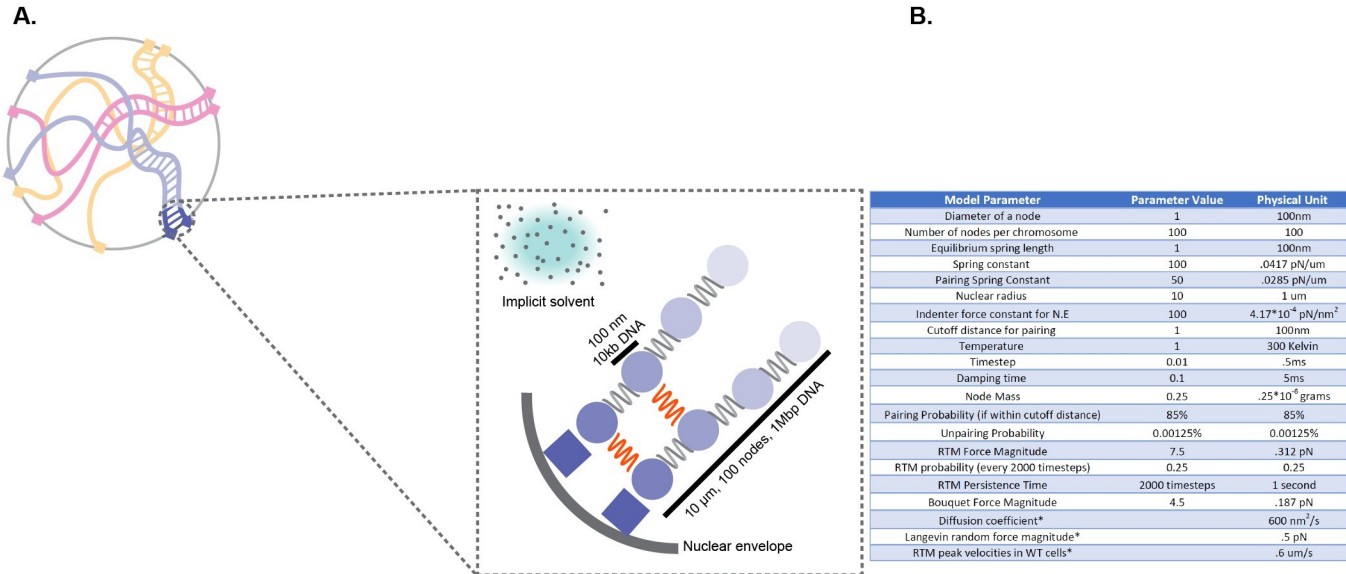

**Fig 2.** Simulating Meiotic Chromosome Pairing (A) To model meiotic chromosome pairing, we use bead spring polymers undergoing Langevin dynamics. (B) Model parameters used in the simulation.

wildtype (WT). This result is inconsistent with the currently available experimental data which shows that entry into anaphase is delayed in the RTM deficient strains [49], presumably due to slower pairing.

The simulation in (**Fig 3A**) assumed that every node could pair with its homolog. In reality, the distribution of pairing sites will be dictated by the distribution of double strand breaks or

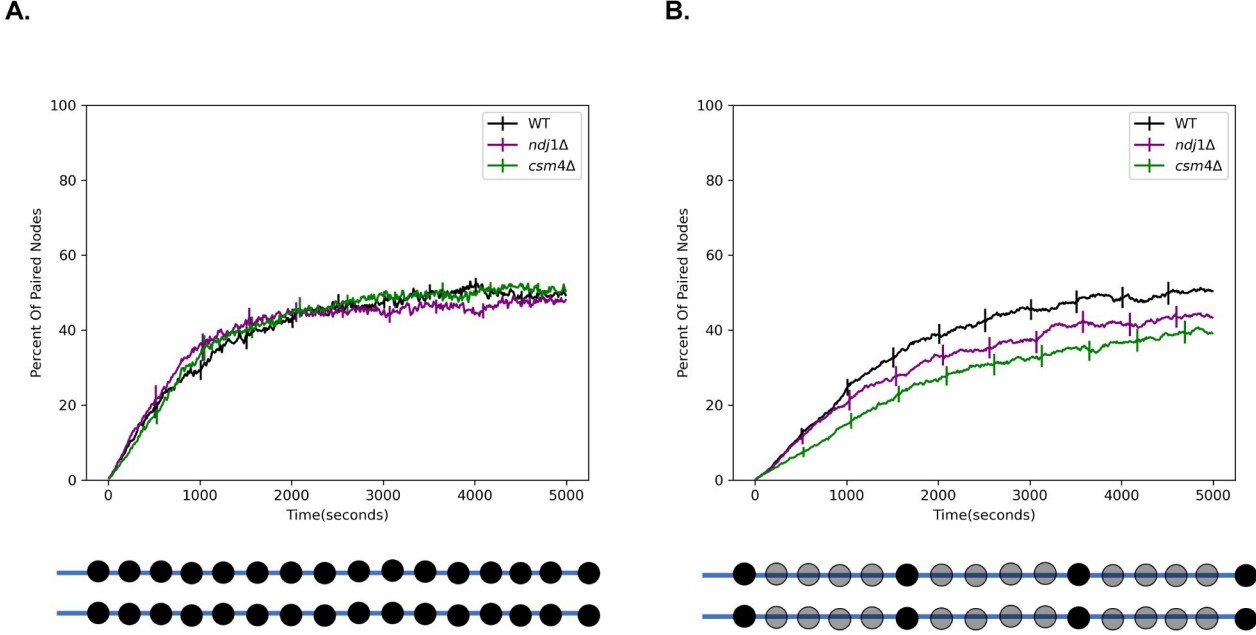

**Fig 3. Rapid telomere movement accelerates homologous chromosome pairing.** (A) Percent of paired nodes vs time for the case that pairing occurs along the entire chromosome. The mutant strains *ndj1Δ* and *csm4Δ* pair with the same kinetics as WT. (B) Percent of paired nodes vs time for the case that pairing sites are spread out along the chromosome. RTMs increase the rate of pairing of wildtype (WT) as compared to the mutant strains. Error bars indicate 95% confidence intervals.

other mediators of pairing. To see if our model would give different results if the density of pairing sites was reduced, we carried out a second set of simulations in which we uniformly space twenty pairing sites along the length of the chromosome. In the case where the pairing sites are spread out along the length of the chromosome, we find that the WT strain pairs the fastest, followed by *ndj1Δ*, followed by *csm4Δ* (**Fig 3B**). This is consistent with experimental data from [49] in which WT is seen to enter anaphase the fastest, followed by *ndj1Δ*, followed by *csm4Δ*.

Thus, using the more realistic distribution of pairing sites, we see that the rapid telomere movements experienced by WT cells do in fact speed up pairing relative to the RTM deficient strains. This result is consistent with the idea that a function of the RTMs is to increase collisions between homologous loci to speed up pairing.

To better understand the differences between these two scenarios we looked at the time course data for individual simulation runs (**S3 and S4 Figs**). We see that in the case that every node can pair progressive zippering dominates the pairing behavior and eliminates differences between individual strains. In the case that pairing sites are spread out, progressive zippering is much weaker which leads to stronger differences between individual strains (& individual runs).

To further highlight the differences in the progressive zippering behavior between these two cases, we plotted the distance between corresponding nodes through time as a heatmap (**S5 and S6 Figs**), as was done in [4,5]. Using this style of plot, fast zippering will appear as a near vertical shift to the color blue when reading the plot from left to right. We see that in the case that every node can pair, fast zippering is occurring in every strain. In the case that pairing sites are spread out, there is a much more gradual shift to blue, indicating that progressive zippering is not as dominant.

## Meiotic bouquet speeds up pairing

RTMs are not the only unusual aspect of meiotic nuclear organization. A meiotic bouquet, in which the ends of the chromosomes cluster to a small region of the nuclear surface early in meiosis (**Fig 4A**), has been observed in a number of different organisms [50,51]. It has been hypothesized that the meiotic bouquet serves to physically align the chromosomes to reduce the search space and increase the pairing rate [52]. Still, the exact role of the meiotic bouquet remains unknown, largely due to the fact that mutants that impair the meiotic bouquet also impair RTM magnitude and frequency [49].

To test whether the meiotic bouquet leads to an increase in pairing rates in our simulation, we ran simulations both in the presence and absence of a bouquet, which we represent as an additional force on the telomeres biasing them to a sub-region of the nuclear surface. We then measure the number of paired nodes over time and plot the resulting time course. We find that the addition of a meiotic bouquet allows the chromosomes to pair faster relative to the case where there is no bouquet (**Fig 4B**). This result is consistent with the idea that the purpose of the meiotic bouquet is to physically align the chromosomes in order to speed up pairing.

## Increase in nuclear size slows down pairing

Before the onset of meiosis, the nuclear volume increases dramatically (Fig 5A), often by a factor of at least two [1,13,53]. In the context of pairing, this increase in nuclear volume seems counterproductive since it gives the chromosomes a much larger search space in which they must find their homolog. In a previous study that used a phantom polymer model [4], we found that an increase in nuclear size reduced the speed at which chromosomes pair. Here, we wanted to test whether this remains true using our updated model which includes excluded

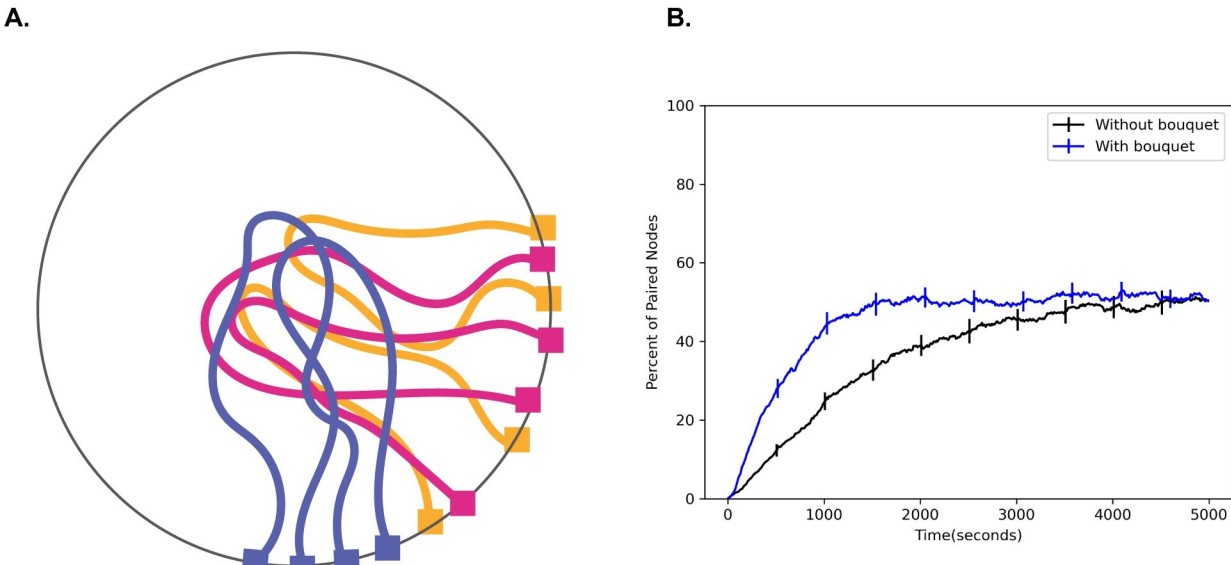

**Fig 4. Effect of bouquet on pairing speed.** (A) Cartoon schematic of the meiotic bouquet, an event in early prophase where the telomeres of chromosomes cluster together. (B) Percent of paired nodes vs. time in the presence and absence of a bouquet shows that pairing is achieved more rapidly in the presence of a bouquet. Error bars indicate 95% confidence intervals.

volume. Our rationale is that excluded volume effects can increase the effective friction in dense polymer networks, so that in principle, it may be advantageous to dilute the polymer mixture by increasing the nuclear volume to drive faster diffusion and thus faster pairing.

To determine whether this is the case, we ran pairing simulations at different nuclear radii (Fig 5B) ranging from 0.8 microns to 1.2 microns and plotted the number of paired nodes vs

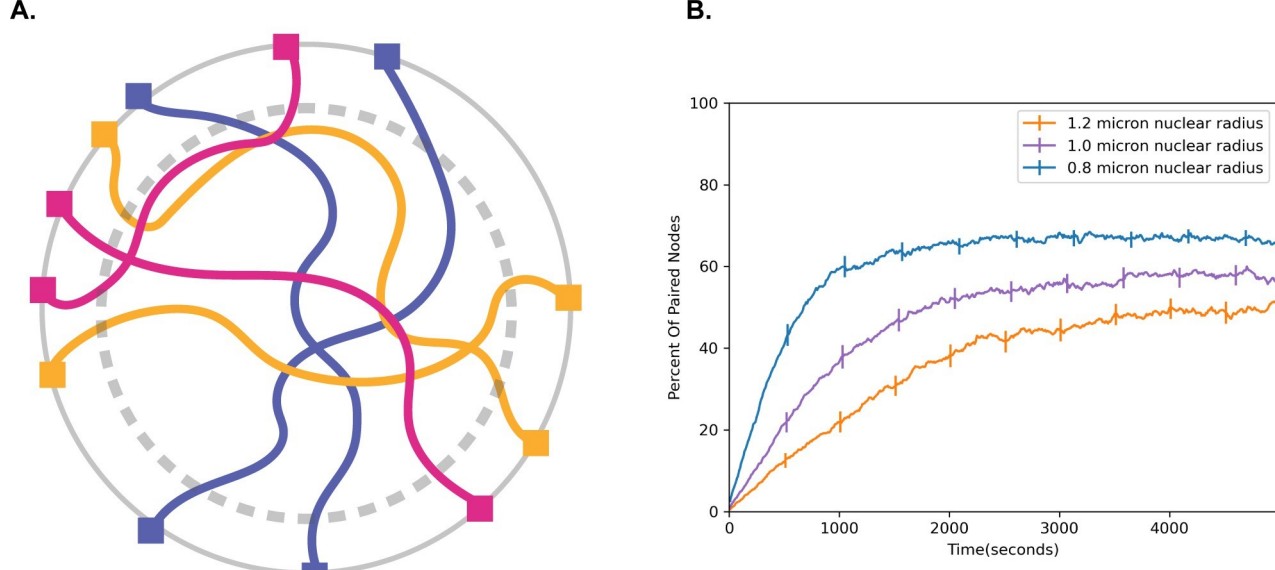

**Fig 5. Effect of nuclear volume increase on pairing speed.** (A) Cartoon schematic of nuclear volume increase. Gray dashed circle represents the original nuclear boundary. (B) Percent of paired nodes vs. time for three nuclear radii shows that pairing is progressively slower as the nuclear radius is increased. Error bars indicate 95% confidence intervals.

time. Consistent with previous simulations, we find that increasing the nuclear size slows down pairing, raising the question of why cells may have evolved to increase nuclear size during meiosis. We will revisit this question below.

## RTMs increase the number of cells with interlocks remaining

Our initial results support the idea that rapid telomere motion functions to speed up pairing. Another proposed function of the rapid telomere movements may be to pull apart already paired regions of the chromosome in order to resolve interlocks (**Fig 1D**). A recent study in the organism *Arabidopsis thaliana* found evidence for this idea [54]. The authors report that the chromosome movement deficient mutant *nup136Δ* had greater numbers of interlocks remaining after synapsis.

With this in mind, we tested whether rapid telomere movements contribute to interlock resolution using our model. First, we noted that the two types of interlocks previously described [54,55], open and closed, are both seen in our simulations (**Fig 6A**). Open interlocks occur when one or both homologous chromosomes become trapped between another pair of homologous chromosomes and thus impede the pairing of a single pair of homologous chromosomes. Closed interlocks occur when the strands of two pairs of homologous chromosomes become trapped in a chain-like structure which prevents both pairs of homologous chromosomes from fully pairing. We then ran simulations of our three in silico strains: WT, *ndj1Δ*, and *csm4Δ*. Since only the WT strain experiences the rapid telomere pulls, we expected that if the RTMs were involved in interlock resolution, then we should see an increase in the percentage of cells with interlocks remaining in the two simulated mutant strains.

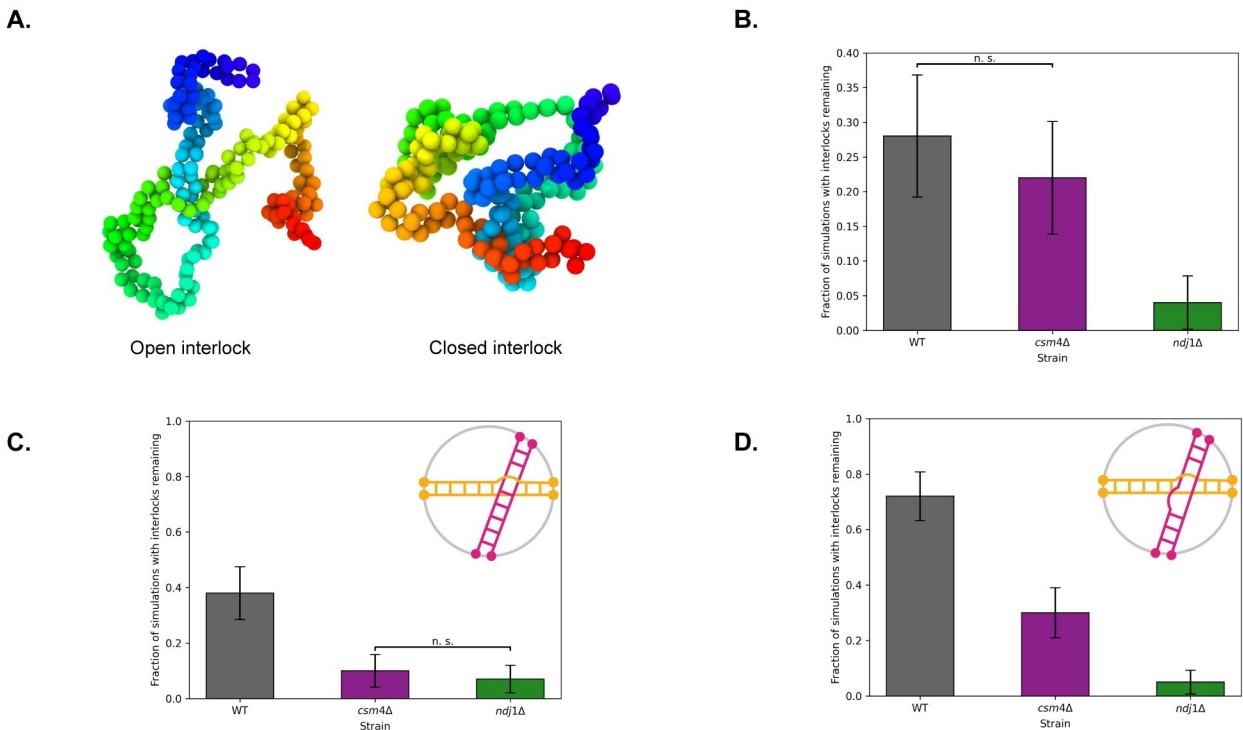

**Fig 6. Role of RTMs on interlocks.** (A) Model captures both types of interlocks that are known to occur experimentally. (B) Loss of RTMs leads to a greater percentage of cells with interlocks. (C) Resolution of Open Interlocks. Graph shows percent of simulations with interlocks remaining after ~ 2 hours in the case that chromosomes start in an open interlock configuration (lower left). (d) Percent of simulations with interlocks remaining after ~ 2 hours in the case that chromosomes start in a closed interlock configuration (lower right). Error bars indicate 95% confidence intervals.

Surprisingly, we see the opposite result, with a greater percentage of the simulated WT cells having interlocks remaining after the completion of pairing relative to the two RTM deficient strains (**Fig 6B**).

The increase in interlocks seen in the presence of RTMs could either mean that interlocks form more readily, or that they fail to be resolved. The simulations described thus far capture both possibilities, because we initialized the chromosomes in an initially unpaired random orientation inside the nucleus, allowed them to pair, then assessed each cell for the existence of an interlock at the end of each simulation. It has been postulated that RTMs could help resolve interlocks by pulling chromosomes away from each other [27,28,54], suggesting that perhaps we were seeing a competition between increased interlock formation and increased interlock resolution, which would predict that in the presence of RTMs, interlocks should resolve more readily once they form.

In order to specifically test the influence of RTMs on interlock resolution, we initialized each simulation in an already paired, interlocked configuration, either open (**Fig 6C**) or closed (**Fig 6D**). These initial configurations were chosen so that the interlock was near the middle of the polymers. These initial configurations were then randomized via an equilibration step that did not allow the bond topology to change (to prevent the interlock from resolving before t = 0). At t = 0 we run the simulation forward in time for two hours and then at the end of the simulation we assess whether the interlock is still there. This experiment tells us about the probability of an interlock resolving in each condition after a set amount of time. In both cases, loss of RTMs leads to a decrease in interlocks, suggesting that RTMs normally impede interlock resolution. The effect was much stronger on closed interlocks.

## Effect of single-end attachment

Our simulations in (**Fig 6**) indicate that fewer interlocks are observed when telomeres are detached from the NE. This result suggests that one way to avoid interlocks is to have detached telomeres, but then the potential benefits of RTMs for accelerating pairing would be lost. In one well studied model system for meiosis, *C. elegans*, only one end of each chromosome is attached to the nuclear envelope (**Fig 7A**) [56], raising the question of whether having one chromosome end attached to the NE to experience RTMs, while the other is detached to avoid interlocks, might be an optimal strategy. We simulated this situation of a single attached end and found a dramatic reduction in the percentage of cells with interlocks in the case that only one end of the chromosomes is attached to the nuclear envelope. (**Fig 7C**). We note that this reduction in interlocks comes at the cost of slightly slower pairing compared to when both ends are attached (**Fig 7B**).

## The meiotic bouquet does not significantly alter the number of cells with interlocks remaining

While we already showed in a previous section that the meiotic bouquet speeds up pairing in our model, we also sought to investigate whether the meiotic bouquet had any effect on the number of cells with interlocked chromosomes at the end of pairing.

To do this, we run two sets of simulations, one with a bouquet, and one without a bouquet. For each case, we run full pairing simulations of pairing and count the number of simulations in which there are interlocks at the end as discussed in the methods section. We find that the meiotic bouquet does not significantly change the percentage of cells in which there are interlocks (**Fig 8A**).

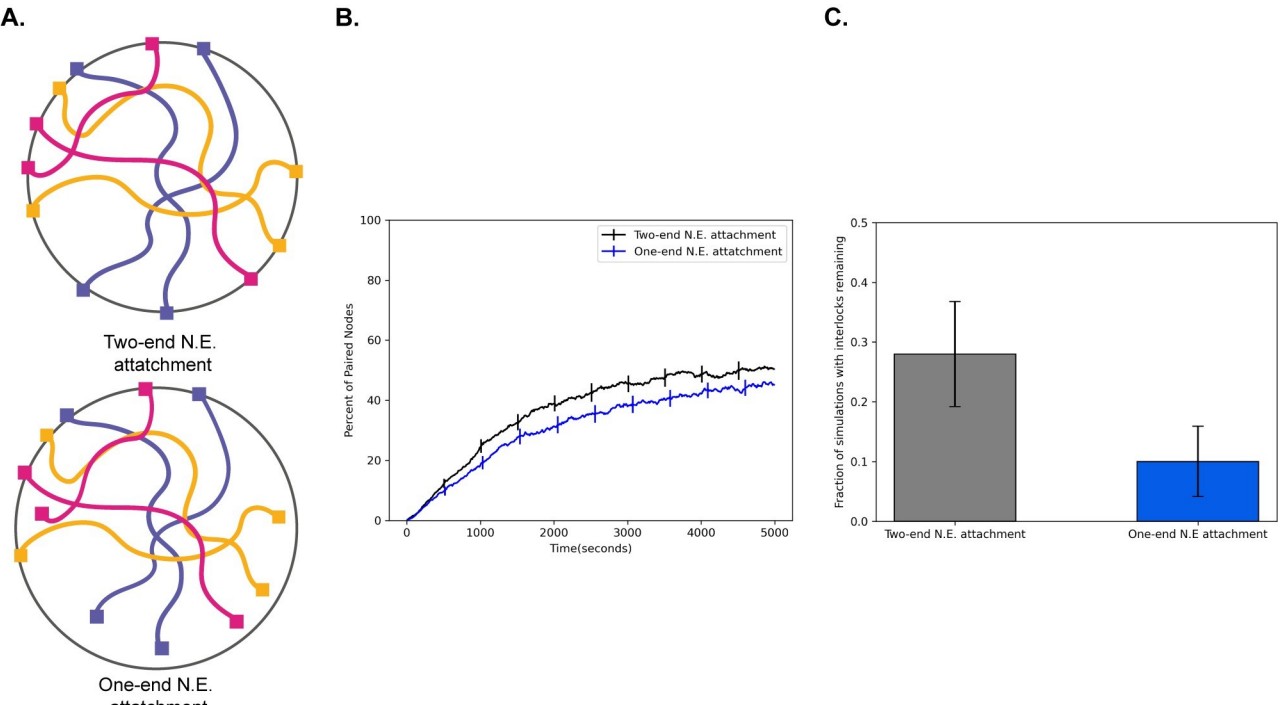

**Fig 7. Single-end attachment facilitates interlock resolution.** (A) Cartoon schematic of double end vs. single end chromosome attachment to the NE (B) Percent of paired nodes vs. time for single end vs. double end chromosome attachment shows an increase in pairing speed with both ends attached. (C) Percent of simulations with interlocks remaining for single end vs double end chromosome attachment shows a greater number of cells with interlocks remaining in the case where both ends are attached to the N.E. Error bars indicate 95% confidence intervals.

## Increasing the nuclear size reduces the number of interlocks

When polymers are densely packed in a finite volume, they become entangled. For example, it is well established that the probability of polymer knotting is a function of the size of the

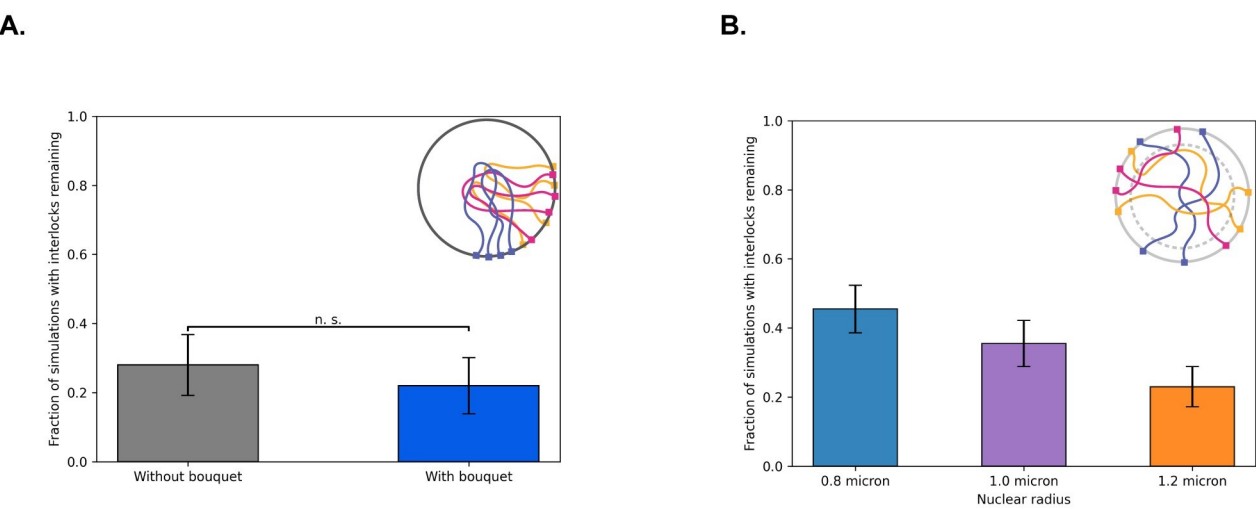

**Fig 8. Effect of bouquet and nuclear volume increase on interlocks.** (a) Percent of simulations with interlocks remaining in the presence and absence of a meiotic bouquet shows that the number of interlocks does not change significantly in the presence of a bouquet. (b) Percent of simulations with interlocks remaining at three different nuclear radii shows that increasing the nuclear volume decreases the number of cells with interlocks. Error bars indicate 95% confidence intervals.

container that the polymer is enclosed in [57]. Studies have found that the probability of knotting sharply increases with increasing confinement in both linear polymers [57], and random ring polymers [58]. These studies suggest that the nuclear volume, in addition to its effects on pairing collisions reported above, might also affect the degree of entanglement between chromosomes, leading to an effect on the number of interlocks that occur. A large nucleus would be predicted to form fewer interlocks during pairing.

To test this prediction, we calculated the number of cells with topological interlocks between chromosomes at the end of the simulation, and found that while larger nuclei lead to slower pairing kinetics, the increased volume does in fact lead to a dramatic reduction in the number of interlocks (**Fig 8B**).

### In the absence of strand passage, interlock resolution proceeds in three sequential steps: interlock migration, followed by telomere unpairing, then by diffusive unwinding

Interlocks are commonly observed in meiosis, but eventually resolve by the pachytene stage [13,14,19,54,55]. Our model recapitulates this formation and resolution of interlocks, but it is not obvious a priori how such interlocks in linear polymers are able to resolve simply from a combination of reversible pairing and random motion, with no mechanisms for strand breakage or for directed untangling. We therefore sought to understand the step-by-step process by which interlocks resolve in our model.

To see exactly how interlocks resolved, we visually studied the trajectories of these interlock resolution events. (**Fig 9A**) shows a representative example of an interlock resolution event which we further simplify in (**Fig 9B**). We note that there is a meiotic bouquet in this example for visual simplicity, but the process is exactly the same in the absence of a bouquet.

We find that interlocks can occur anywhere along the length of the chromosome, but because the ends are tethered to the nuclear envelope, the chromosome endpoints are the only places an interlock can resolve (**Fig 9C**). This means the interlock must first migrate towards the telomeres on the NE (**Fig 9C**), where it is then held in a semi-stable state by the large number of paired nodes on one side. To fully resolve an interlock, two additional steps must sequentially take place. First, the telomeric regions of the paired chromosomes must unpair (**Fig 9C**). This presents an opportunity for interlock resolution. Next, the telomeres of the interlocked strand must diffusively unwind itself around the other telomeres for the interlock to fully resolve. Typically, interlock resolution is then followed by rapid zippering of the remaining unpaired strands (**Fig 9C**) making the disentanglement effectively an irreversible process.

### Timescale estimates of the three steps in interlock resolution suggest that interlock migration is the limiting step in the process

To estimate the timescales of the different interlock resolution steps, we visually analyzed the trajectories of fifteen interlock resolution events using the software program OVITO. For each interlock resolution event we noted the time it took to complete the three steps in interlock resolution: 1) Time to interlock migration, 2) Time to telomeric unpairing, and 3) Time to diffusive unwinding.

We display these results as a table in the supporting information (**S7 Fig**). We find that interlock migration is the limiting step in this process, taking an average of ~2850 seconds to complete, followed by diffusive unwinding at ~720 seconds, and telomeric unpairing at ~270 seconds.

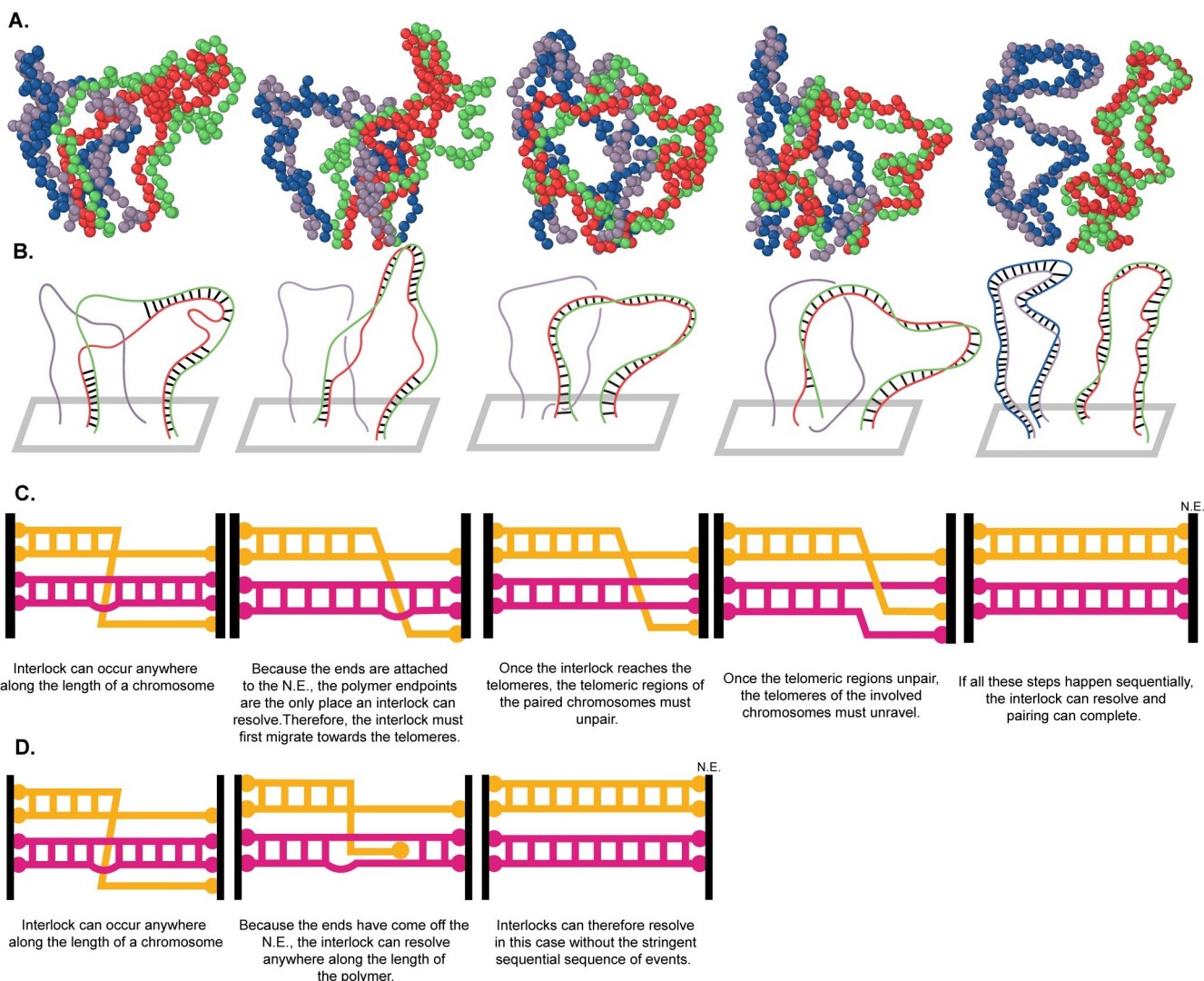

**Fig 9. Geometry of interlock resolution.** A) Simulation results showing interlock migration to telomeres followed by complete resolution. (B) Simplified drawing of the same interlock resolution event showing gradual migration of the interlock to the end, after which telomere re-arrangement leads to the blue chromosome becoming disentangled from the red/green paired chromosome. (C) Cartoons of steps required for interlock resolution with both ends are attached to the NE (D) Cartoon representation of steps required for interlock resolution if at least one end of the chromosomes is unattached.

## Increasing the degree of entanglement leads to more interlocks

The number of interlocks is likely to depend strongly on the degree of entanglement. There are two ways to easily modulate the degree of entanglement in our model 1) change the volume of confinement, and 2) change the number of nodes per polymer chain.

First, we sought to quantify the number of entanglements per polymer chain for three different nuclear radii. To do this, we used primitive path analysis as discussed in the methods section. We find that our polymers contain between 0–4 entanglements per polymer, with an increasing number of entanglements as you decrease the nuclear radius (S8 Fig). In (Fig 8B) we also showed that increasing the nuclear radius decreases the number of interlocks. Together these results support the idea that increasing the degree of entanglement leads to more interlocks.

Another way to modulate the degree of entanglement is to modulate the number of nodes per polymer chain, as longer polymers will be more highly entangled [23]. To test whether using longer polymers leads to more interlocks, we ran pairing simulations at three polymer lengths (N = 80, N = 100, and N = 120 nodes per chain) and evaluated them at the end for the existence of interlocks (**S9 Fig**). We find that the number of interlocks increases as you increase the number of nodes per polymer chain, which supports the idea that the number of interlocks depends on the degree of entanglement.

## Discussion

### In the absence of strand passage, attachment of both ends of the telomeres to the NE means interlock resolution can only occur at the polymer endpoints

One important realization from our model is that if both ends of the chromosomes are tethered to the nuclear envelope, then interlock resolution of the form we described can only occur at the chromosome endpoints. This realization came, in part, from looking at diagrams describing the resolution of knots in a simplified model of knot formation described in [59]. In this study, the authors look at the probability of knot formation in a system that consists of linear polymers confined to move inside of a confining volume. They note that knot formation and resolution of knots occurs when a single polymer end weaves through a parallel section of the surrounding strands, a move the authors call a "braid move". We note that in our system, the polymers are under an additional constraint that makes a braid move much more difficult: namely, that the ends of the polymers are attached to the confining surface. In this case, a polymer end cannot weave through the surrounding strands, instead a braid move can only occur if at least *two* polymer endpoints weave around *each other*, an important distinction.

The biological implication of this is that in organisms where both ends of the chromosome are tethered to the NE, an interlock that resolves in this way must do so near the nuclear envelope.

### Can topoisomerases catalyze strand passage during meiosis?

One possible way to resolve interlocks could be to pass the chromatin strands through each other. In other parts of the cell cycle when chromosomes consist of a single double strand of DNA this reaction is easily catalyzed by topoisomerases such as topo II, which are known to temporarily cut double stranded DNA while allowing a strand to pass another strand through the resulting gap [60]. While it is true that topoisomerases are upregulated during meiosis [61], at this stage of the cell cycle each chromosome consists of two sister chromatids which are bound together along their length. In order to pass strands through each other then, a topoisomerase would have to pass two double helices through another pair of double helices, which to our knowledge is not a known activity of topoisomerases. The function of topoisomerase may be irrelevant anyway given that the chromosomes are organized around a filamentous protein axial element, which is not a substrate for breakage by topo II. While it has been reported that topoisomerase plays a role in the resolution of interlocks [54], it cannot be ruled out that this reflects an indirect effect of topo II, perhaps a result of chromatin reorganization, rather than from the direct resolution of interlocks via strand passage. In fact, a recent paper shows that topo II is required for meiotic double stranded break repair progression, where it is thought to relieve stress caused by supercoiling [62]. As discussed in the introduction, the presence of an axial element during pairing would further prevent passage of one chromosome through another.

In light of these problems with the idea of topoisomerase catalyzed interlock resolution, an important result of our model is that under the physical constraints experienced by meiotic chromosomes, interlock resolution can occur just from reversible pairing coupled with random motion.

## Is the detachment of the telomeres from the NE a strategy for interlock resolution in any organism?

Our model shows that it is possible to resolve interlocks without any strand passage catalyzed by enzymatic activity, and that this process can be dramatically enhanced if one or both ends of the chromosomes come unattached from the nuclear envelope. Some organisms such as *C. elegans* only have one end of the chromosomes attached by default. Our results suggest that this configuration may help to avoid interlocks but could lead to slower pairing. However, given that *C. elegans* carries out pairing via dedicated pairing centers located close to the attachment point [56,63,64], RTMs applied at the attached end could be sufficient to speed pairing, and increased motion of the other end, far from the pairing center, would be less relevant anyway.

The large effect on interlock reduction of chromosome end detachment raises the question of whether in some organisms, in which both ends of the telomeres are typically attached to the NE, transient detachment might still occur in order to resolve interlocks. In support of this theory, it has been reported [19] that in lily meiocytes 14 out of 48 telomeres were not associated with the NE at early zygotene. While *lilium* is no longer a widely used organism in the study of meiotic pairing, our results coupled with this historical observation motivate future experiments in which the telomeric ends are closely monitored for transient departure from the NE. Given that in most systems, the telomeres appear to be attached to the NE almost all of the time, it is not likely that the detachment would last long enough for the chromosomes to undergo extensive long-range motion such as reptation. Instead, we imagine that a transient detachment could provide a rapid way to carry out the final step in interlock resolution, once the interlock has migrated near the NE, by allowing the captured chromosome to escape the interlock (**Fig 9D**).

## Why is *csm4Δ* a more deleterious mutation than *ndj1Δ*?

A perhaps surprising result from genetic experiments is the fact that *csm4Δ* is a more deleterious mutation than *ndj1Δ*, both in the time required for entry into anaphase [49] and in terms of spore viability [35], which is opposite to the order of their phenotypic effects on chromosome organization. While both mutations disable the rapid telomere pulls, in the *ndj1Δ* strain the ends of the chromosomes come completely detached from the nuclear envelope, while in the *csm4Δ* strain the ends remain attached. *ndj1Δ* would appear to be the more dramatic mutation. Why then is the *ndj1Δ* mutant more viable than *csm4Δ*?

A key methodological feature of our model is the representation of these strains in silico, which was accomplished by matching the velocities of our telomeres to the experimentally measured velocities of the telomeric ends. While building the model we noted that the *csm4Δ* strain has a much lower average telomere velocity than the WT cell. It appears that in the absence of the rapid telomere pulls, the ends of the chromosomes are effectively tethered to a particular place on the nuclear envelope, from which they diffuse very slowly. This extreme tethering of the telomeric ends to the nuclear envelope could in principle highly constrain the search process for homologous pairing, thereby impeding the completion of homologous pairing and synapsis. To this effect, we see in (**S3 Fig**) that *csm4Δ* cells typically have a much slower ascent towards full pairing than *ndj1Δ*.

This observation sheds light on why (in the absence of RTMs) it might actually be better to completely detach the chromosome ends from the nuclear envelope. Once the telomeres are detached from the NE in the *ndj1Δ* strain, the telomeric ends are free to diffuse throughout the nucleus at the same diffusion coefficient of meiotic chromatin, thereby releasing the spatial constraints on the homologous search process.

## Comparison to published data on mutant pairing kinetics

In [32] the homolog pairing kinetics were experimentally compared in WT and *ndj1Δ* cells. This was accomplished by tagging a single locus on each homologous chromosome and measuring the fraction of time the spots appear colocalized. Based on visual inspection of their figure, WT reaches a plateau that is approximately 10% higher than *ndj1Δ*, which is comparable to the difference we see (**Fig 3B**) which is also approximately 10%. In the same experimental study, it was found that WT cells reach half maximum pairing approximately ~ 15% faster than *ndj1Δ* cells, which is comparable to our result. Looking at our data in (**Fig 3B**), our simulated WT cells reach half maximum approximately ~19% faster than our simulated *ndj1Δ* cells.

## Comparison to published data on interlock resolution

In Martinez-Garcia et al [54], they show that an RTM deficient A. *Thaliana* mutant, *nup136Δ* has more interlocks than the wild-type cells. Importantly, this result seems to contradict the conclusions presented in the current study. Indeed, our simulated RTM deficient yeast strains contain less interlocks than in the WT cells. We would like to speculate on a possibility for this apparent contradiction.

First, we note there are well established differences between the canonically studied RTM deficient budding yeast strains *csm4Δ* and *ndj1Δ*. Experimentally, it is well established that *csm4Δ* strain is more deleterious than *ndj1Δ*, both in terms of time it takes to enter anaphase, as well as spore viability. The simulation study presented here supports these results: we see that *csm4Δ* is both slower at pairing and has more interlocks than *ndj1Δ*. Importantly, these differences arise from a key difference between these two mutants (which we built into our model): the behavior of the telomeres. In *ndj1Δ* the telomeres detach from the nuclear envelope, while in *csm4Δ* they remain attached. Understanding the behavior of the telomeres is thus key to understanding the physical effect of these mutations. While both achieve the goal of turning off the RTMs, they do it in two very different ways which results in very different physical situations.

To our knowledge, there is no published study that carefully examines the behavior of the telomeres in *nup136Δ*. Additionally, *nup136 Δ* has been shown to alter the nuclear morphology [65], going from highly elongated in WT cells, to highly spherical in the mutant strain, an effect not seen in either *ndj1Δ* or *csm4Δ*. There is thus no true contradiction. Each RTM deficient strain is unique and careful attention to detail is required to model specific strains. There is unfortunately currently not enough experimental data on the velocity and localization of the telomeres in *nup136Δ* to model this strain. Perhaps a future version of the model will incorporate future data to better understand interlock resolution in this strain.

## Coarse-grained molecular dynamics modeling of meiosis

The inspiration for our model came from an earlier model [4,5] where we used a similar bead spring polymer model to study meiotic chromosome pairing. This model represents an update on the previous model with two important improvements: 1) We created this model using LAMMPS, a popular framework for building molecular dynamics models. LAMMPS is open source and maintained by an active academic community which provides pre-built executables

that are easy to install. This means anyone can run our simulations using the included initialization files. 2) This updated model includes excluded volume, which means polymers cannot pass through each other. This is what allowed us to be able to study the resolution of interlocks, something that would not have been possible using our previous model.

While there are other physical models of meiotic pairing, our model is unique in that it attempts to capture the large-scale cell biological picture of meiotic pairing including the random initial 3D search, the rapid telomere motion, and additional physical constraints such as telomere attachment to the NE.

A recent model [62] for example, is focused on somatic pairing in *Drosophila* rather than meiotic pairing. The authors use a kinetic Monte Carlo scheme to show that a "button-based" pairing mechanism in which the pairing sites are spread out along the length of the chromosome can recapitulate chromosome-wide pairing. This is consistent with our previous results that widely spaced pairing sites function effectively to promote pairing [4], which we have now found also applies in our more realistic model with excluded volume (**Fig 3B**). Because the authors were modelling somatic pairing rather than meiotic pairing, their model does not include the meiosis specific constraints like our model.

Another recent model [6], is focused on capturing the signature of physical confinement in loci along paired chromosomes. They thus employ a Rouse polymer model where pairing along the polymer chains occurs via the forced creation of random dynamic linkages, rather than from a search process as it occurs in our model.

Another model [3] is focused on capturing the dramatic horsetail movements observed in S. *pombe*. In this study, the authors focus on the prealignment and sorting of simulated chromosomes, and thus their model does not include physical linking of paired loci.

A model [2] used Markov chain Monte Carlo sampling to show that the meiotic bouquet led to homologous juxtaposition, a result that is consistent with our current results which show that the meiotic bouquet physically aligns the chromosomes to speed up pairing (**Fig 4B**). Their model however, did not include excluded volume interactions or physical linking of pairing loci.

An early model [66] used a cellular automata model to simulate how various aspects of meiotic pairing affected the homology search process. This model included NE attachment and telomere clustering (bouquet formation) but did not include the potential for interlocking or entanglement.

## Supporting information

**S1 Fig. Mean square displacement for evaluating the diffusion coefficient.** (A) We take the slope of the linear portion of the MSD plot and divide it by 6 to get the diffusion coefficient. We note that to avoid confinement effects which would obscure the linear region of the plot, we did this simulation in a rectangular box of equal volume to the nuclear sphere, then used periodic boundary effects to avoid interactions with the simulation box while keeping the density constant.
(TIF)

**S2 Fig. Velocity vs time of a single telomere experiencing active pulls.** (A) Velocity vs time profile shows discrete pulling events where telomeres reach peak velocities of approximately .6 μm/s. In between pulls there are dwell times lasting several seconds.
(TIF)

**S3 Fig. Time course data for individual simulation runs in the case that every node can pair.** Progressive zippering leads to very fast pairing behavior in every strain which eliminates

differences between strains.
(TIF)

**S4 Fig. Time course data for individual simulation runs in the case that pairing sites are spread out.** Progressive zippering is not as strong, and differences between the individual strains (& individual runs) are more apparent.
(TIF)

**S5 Fig. Kymographs for individual simulation runs in the case that every node can pair.** Fast progressive occurs in every strain (seen here as a sudden near vertical line of blue when reading the plot from left to right), which eliminates differences between strains.
(TIF)

**S6 Fig. Kymographs for individual simulation runs in the case that pairing sites are spread out.** Progressive zippering is much slower (seen here as a more gradual shift to blue when reading the plot from left to right), which allows differences between individual strains to be more apparent.
(TIF)

**S7 Fig. Timescale estimates of the three steps in interlock resolution.** Time it takes to complete each step in interlock resolution for fifteen interlock resolution events. We see that interlock migration is the limiting step, followed by diffusive unwinding, then telomeric unpairing.
(TIF)

**S8 Fig. Primitive path analysis to count entanglements.** (A) Representative example of primitive path analysis shows polymer relaxing to its primitive path which shows N = 2 kinks at places of entanglement. (B) Table documenting the number of entanglements for fifteen different simulation runs at three nuclear radii (45 simulations total).
(TIF)

**S9 Fig. Effect of polymer length on interlocks: We ran pairing simulations at three polymer lengths and evaluated them at the end for the existence of interlocks.** We find that increasing the number of nodes increases the number of interlocks.
(TIF)

## Acknowledgments

We thank members of the Fung and Marshall labs for many helpful discussions, particularly Beth Rockmill and Mike Pollard.

## Author Contributions

**Conceptualization:** Erik J. Navarro, Wallace F. Marshall, Jennifer C. Fung.

**Funding acquisition:** Wallace F. Marshall, Jennifer C. Fung.

**Investigation:** Erik J. Navarro.

**Methodology:** Erik J. Navarro.

**Supervision:** Wallace F. Marshall, Jennifer C. Fung.

**Validation:** Erik J. Navarro.

**Writing – original draft:** Erik J. Navarro.

**Writing – review & editing:** Erik J. Navarro, Wallace F. Marshall, Jennifer C. Fung.

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
