## [Decision Letter · Decision Letter 0]

11 Feb 2022

Dear Dr. Fung,

Thank you very much for submitting your manuscript "Modeling cell biological features of meiotic chromosome pairing" for consideration at PLOS Computational Biology.

As with all papers reviewed by the journal, your manuscript was reviewed by members of the editorial board and by several independent reviewers. In light of the reviews (below this email), we would like to invite the resubmission of a significantly-revised version that takes into account the reviewers' comments.

We cannot make any decision about publication until we have seen the revised manuscript and your response to the reviewers' comments. Your revised manuscript is also likely to be sent to reviewers for further evaluation.

Sincerely,

Attila Csikász-Nagy

Associate Editor

PLOS Computational Biology

Arne Elofsson

Deputy Editor

PLOS Computational Biology

Reviewer's Responses to Questions

**Comments to the Authors:**

Reviewer #1: This is an exciting new computational study of the

homologous chromosome pairing in meiosis. The authors simulate

this process in 3d and without ignoring entanglements,

which previous models did not do. (There is an exemplary

detailed comparison with previous models in the Discussion).

The authors use Langevin dynamics of polymer chains representing chromosome strands with

telomeres moving rapidly on the nuclear envelope to simulate the spatial-temporal

dynamics of chromosomes with emphasis on speed and accuracy of pairing. They find that

function of the RTMs is to increase collisions between homologous loci to speed up pairing;

that addition of a meiotic bouquet allows the chromosomes to pair faster and does not

significantly alter the number of cells with interlocks remaining; that

increase in nuclear size slows down pairing and reduces the number of interlocks.

The surprising result is that RTMs increase the number of cells with interlocks remaining.

The most interesting result, I think, is computational demonstration that a dramatic

reduction in the percentage of cells with interlocks occurs in the case when only one

end of the chromosomes is attached to the nuclear envelope. What I also liked a lot

is that the model vividly demonstrates the sequence of the interlock resolution: a) interlock

migration, b) telomere unpairing, c) diffusive unwinding.

The paper is, with one exception (see below), well written, novel, significant.

Major problem:

Model equations in the Methods:

- number the equations

- the equations' description is incredibly sloppy, bordering

on disrespect to the reader... What are zeta, m, etc? What is

the physics behind the equations? m can't be mass , can it?? etc etc

- are the nodes of chromosomes jump between the lattice points or move

in 3D continuous space?

- what is the 'indenter'? Why is the spring force proportional to the

square of of the deformation? (why not use Hookean linear spring)?

why is the spring constant in pN/nm then? Dimensions do not make sense...

- why ten million time steps? what does it correspond to in real time?

- telomers' velocity - how is it directed? randomly? complete randomization

at every time step? and more, and more... tens of questions...

- there are some vague hints to what the meaning of the parameters is

in "Damping Time & Temperature", but I still do not understand any of

it, despite my education in Physics. Come on guys! Explain what you

are doing. Re-write the Methods completely keeping in mind: say you

want a student use your method. Explain to a student who never heard it

before...

Two minor problems:

Intro: not exactly clear what 'interlock' is;

it is hard to understand from fig 1d how do telomers'

movements 'resolve interlocks'

Results:

fig 3b - speed of pairing may be statistically different in WT and mutants,

but practically, the difference is insignificant, no? I see ~ 15% difference,

is that right? What kind of difference do people see experimentally? Probably

much greater?

Reviewer #2: The manuscript describes a polymer model to study meiotic chromosome pairing. It systematically analyzes the role of specific mechanisms on the pairing efficiency and on the resolution of entanglements.

Compared to their previous works (Marshall & Fung, 2016 and 2019), authors introduce self-avoidance in their model to investigate the formation and resolution of the so-called interlocks. Then, they discuss their results in light of the experimental knowledge on meiotic pairing.

Overall, the paper is well written and easy to read, and addressed, for the first time in such biological context, the question of interlocks. However, there are several limitations that need to be addressed.

Main issues:

1) The study is purely computational. While it is not a problem per se, it would really enhance the impact of the work if some quantitative or qualitative comparison is done with quantitative experimental data.

2) It is not very clear what the first 3 parts of the Results section (on pairing dynamics) bring compared to the previous works performed by the authors. Conclusions seems rather similar.

3) The main originality came from the study of interlocks. So, it might be nice to refer to the most original results in the title of the paper. Moreover, the term ‘interlock’ is directly used in the introduction and in the rest of the paper without precisely defining what is an ‘interlock’ (same for open or closed interlocks).

4) Overall, I found the paper very descriptive (with a nice analysis of the effects of many parameters & mechanisms) and it will gain in impact if more insights into the physics of the system were given. For example, why WT, ndj1Delta, csm4Delta have similar dynamics in Fig.3A but a different in Fig.3B? why ndj1Delta is more efficient than csm4Delta (Fig.3B)? what is the role of the interplay between zippering and inter-chromosome dynamics?

5) The interlock statistics and dynamics is likely to depend strongly on the initial configurations of the polymers. This is an important point that need to be studied. Experimentally, what is the know initial organization? Rabl?

6) The section on interlock resolution would be really improved if some quantifications can be done. For example, estimation of the time-scales of the different steps might be helpful.

7) The Methods section is sometimes unclear and confusing.

a. ‘F’ are forces but equations describes sometimes a force indeed but sometimes a potential

b. The best potential to avoid chain crossing is FENE while authors use spring+LJ

c. The construction of the initial configurations should be better described (are all the polymers initialize simultaneously? what are their relative orientations? How the constraints for telomeres are managed?...)

d. How the polymers are confined inside a sphere is not clear. Is F (L113) a force? A potential? What is an indenter?

e. How the persistent forces on telomeres and the bouquet (size of the bouquet ?) are implemented?

f. Motivations behind the choices of spring constants are not given.

g. Regarding time mapping: authors compare the predicted MSD to experimental observations but do not show a figure. Moreover, for a polymer, MSD scales as t^(1/2) so the unit of effective diffusion coefficient is in [length]^2/[time]^(1/2) and not [length]^2/[time] as for normal diffusion. More recent estimation of chromatin dynamics in yeast may be found in Hajjoul et al, Genome Research 2013.

Minor issues:

1) Entanglements and knots have been discussed also a lot in the context of interphasic chromosome (eg, crumpled polymers). Might be interesting to compare with authors’ results.

2) Authors rightly associate the formation of interlocks to entanglement, it might be interesting to discuss this a bit more (what is the entanglement length of the simulated polymer?) Long polymers are more sensitive to entanglement, what authors’ conclusions depend on the size of the polymer they choose?

3) References are cited as (Name Date) format but in the reference list, they are numbered.

4) L93: Leonard-> Lennard

5) L106 : ‘matched theoretical predictions’: which one ?

6) For some more recent estimations of the chromatin persistence length in yeast, authors may cite [Arbona et al, Genome Biology 2017; Socol et al , NAR, 2019].

7) L335: “it has been postulated that….”, any reference?

8) L340: how the initial open & closed interlock configurations were chosen?

9) L359: ‘slightly slower pairing’: the slowdown is as for ndj1Delta

10) Panel 9D is not cited in the text

11) L395-401: maybe delocalized to the Method section

12) In the Discussion section, authors should discuss more clearly the contradictions of their results with results from Martinez-Garcia et al, 2018

13) L554-L558: many polymer models of chromosome organization are using similar formalism, so I am not sure that this paragraph is really useful.

**Have the authors made all data and (if applicable) computational code underlying the findings in their manuscript fully available?**

Reviewer #1: Yes

Reviewer #2: Yes

PLOS authors have the option to publish the peer review history of their article (what does this mean?). If published, this will include your full peer review and any attached files.

Reviewer #1: No

Reviewer #2: No
---

## [Decision Letter · Decision Letter 1]

27 Apr 2022

Dear Dr. Fung,

Thank you very much for submitting your manuscript "Modeling cell biological features of meiotic chromosome pairing to study interlock resolution" for consideration at PLOS Computational Biology. As with all papers reviewed by the journal, your manuscript was reviewed by members of the editorial board and by several independent reviewers. The reviewers appreciated the attention to an important topic. Based on the reviews, we are likely to accept this manuscript for publication, providing that you modify the manuscript according to the review recommendations.

Please follow the suggestions by referee 2.

Sincerely,

Attila Csikász-Nagy

Associate Editor

PLOS Computational Biology

Arne Elofsson

Deputy Editor

PLOS Computational Biology

[LINK]

Reviewer's Responses to Questions

**Comments to the Authors:**

Reviewer #1: I am satisfied with the revisions

Reviewer #2: The authors addressed most of my previous concerns and the manuscript was greatly improved.

However I still have minor comments :

- the methods part still needs to be more detailed to allow reproducibility:

o the expression in Eq.2 is the standard deviation of the normal distribution used to simulate the stochastic force, it is not the expression of the force itself

o in Eq 4, r -> r_{i,i+1}

o in Eq.5, r-> r_{i,j} and \\in -> \\epsilon

o in Eq.2, K_b-> k_b

o Eq.6: K is different from Eq.4 I guess, please use different notation, same for 'r'. Moreover looking at the LAMMPS page for 'indenter', Eq. 6 is valid only for r<r. moreover=""> o for the RTM, what is the shape and amplitude of the active pulling force ? same for the constant acceleration for the bouquet ?

o to model the pairing, two nodes are paired with a harmonic spring (what is the value of the spring constant ?

o how the pairing parameters (binding/unbinding rate) were chosen ?

o authors claimed they adjust the pulling forces to have speed of 0.6 um/s. how was it done ? any Sup Fig to illustrate that fit ?

o it is unclear how the telomere tethering at the nuclear periphery is forced at the initiation stage (when the initial configuration is built).

- It is not obvious to see on Fig.S2 & S3 that zippering is dominant or weaker. Maybe a representation of individual trajectories as kymographs (as in Marshall & Fung, 2016) might be more explicit.

- put Arabidopsis in italic.</r.>

**Have the authors made all data and (if applicable) computational code underlying the findings in their manuscript fully available?**

Reviewer #1: Yes

Reviewer #2: Yes

PLOS authors have the option to publish the peer review history of their article (what does this mean?). If published, this will include your full peer review and any attached files.

Reviewer #1: No

Reviewer #2: No

Figure Files:

Data Requirements:

Reproducibility:

References:

---

## [Editor Report · Decision Letter 2]

25 May 2022

Dear Dr. Fung,

We are pleased to inform you that your manuscript 'Modeling cell biological features of meiotic chromosome pairing to study interlock resolution' has been provisionally accepted for publication in PLOS Computational Biology.

Best regards,

Attila Csikász-Nagy

Associate Editor

PLOS Computational Biology

Arne Elofsson

Deputy Editor

PLOS Computational Biology

---

## [Editor Report · Acceptance letter]

6 Jun 2022

PCOMPBIOL-D-22-00033R2 

Modeling cell biological features of meiotic chromosome pairing to study interlock resolution

Dear Dr Fung,

I am pleased to inform you that your manuscript has been formally accepted for publication in PLOS Computational Biology. Your manuscript is now with our production department and you will be notified of the publication date in due course.

With kind regards,

Zsofia Freund
